# Ballistic Impact Resistance of UHPC Plates Made with Hybrid Fibers and Low Binder Content

Paulo Rodrigo Dapper [1,*], Hinoel Zamis Ehrendring [1], Fernanda Pacheco [1], Roberto Christ [1,2], Giovanna Costella Menegussi [1], Maria Fernanda de Oliveira [1] and Bernardo Fonseca Tutikian [1]

1   Polytechnical School, UNISINOS University, São Leopoldo 93040-230, Brazil; hzamis@unisinos.br (H.Z.E.); fepacheco@unisinos.br (F.P.); rchrist@unisinos.br (R.C.); giovanna_menegussi@hotmail.com (G.C.M.); mariaon@unisinos.br (M.F.d.O.); bftutikian@unisinos.br (B.F.T.)
2   Department of Civil and Environmental, Universidad de la Costa, Calle 58 #55-66, Barranquilla 080002, Colombia
*   Correspondence: paulo.dapper@icloud.com

**Abstract:** This study assesses the ballistic impact strength of thin plates made of ultra-high-performance concrete (UHPC) with low cement content ($250 \, \text{kg/m}^3$) and volumes of 80% steel and 20% polypropylene (PP) hybrid fibers. The plates were prepared with thicknesses of 30, 50, and 70 mm and fiber volume ratios of 1.5% and 3.0%. Compressive strength, flexural tensile strength, residual strength, and ballistic impact strength were determined using experimental methods. Test results showed that regardless of fiber content, the UHPC specimens prepared with the hybrid fibers showed similar performance against ballistic impact, exerting relatively low impact energy below 1000 J. The UHPC3.0 mixture made with 3.0% hybrid fiber content exhibited the best performance in terms of energy absorption and spalling resistance at impact energy levels greater than 4000 J. Plate sections with thicknesses of 7 mm showed class III performance (highest level), as recommended for military-based applications.

**Keywords:** sustainability; composite materials; impact; structural elements

## 1. Introduction

Civil construction represents one of the largest sectors in the economy. It is a major cause of environmental damage [1–4] as it involves a high consumption of non-renewable materials, the generation of waste, and low material durability [5,6]. In addition, the production of concrete with non-renewable materials implies high levels of cement consumption, with large volumes of carbon dioxide being generated during production [7]. The use of more durable reinforced concrete structures with optimized materials will lead to greater sustainability in the civil construction sector [8]. In this scenario, one objective is the development of alternatives which use the same non-renewable materials but with greater durability and reduced environmental impact. Ultra-high-performance concrete (UHPC) is one of these materials [9], as it allows the optimization of materials and can provide more durable structures.

Ultra-high-performance concrete (UHPC) is a novel construction material that has gained greater acceptance in the construction industry given its superior structural behavior and durability [10–13]. The mechanical characteristics of UHPC exceed those of high-performance concrete. UHPC can show compressive strength well above 150 MPa and direct tensile strength above 100 MPa [14]. With greater mechanical properties (including residual strength after cracking and toughness), UHPC can be used to protect against impact and explosion. Greater protection and safety against attacks from ballistic projectiles is a primary design criterion for public utility buildings and other public agencies [15]. Therefore, research dealing with the use of UHPC for ballistic protection has become increasingly relevant and focuses on the possibility of using novel and resilient materials

in the civil construction industry [16]. UHPC is normally proportioned with high binder content, ranging from 1100 to 1300 kg/m$^3$ [17]. Replacing cement with pozzolans can enhance the rheological performance, shrinkage, cost, and mechanical properties of UHPC, while reducing the environmental impact [18]. According to Meng et al. [19] it is possible to include high volumes of supplementary cementitious materials (SMCs) in UHPC without affecting these characteristics. In this paper, sand gradation was optimized using the modified Andreasen and Andersen packing model (A-A model) to achieve maximum packing density. The results of [19] indicate that optimized UHPC can lead to a 28-day compressive strength of 125 MPa under standard curing conditions and compressive strength of 168–178 MPa with heat curing for 24 h. Such mixtures have a unit cost per compressive strength at 28 days of \$4.1–4.5/m$^3$/MPa under standard curing. Substitution of 40–60% of cement with fly ash can enhance workability, reduce water consumption, and improve cracking behavior due to the reduction of hydration heat [20].

The superior tensile and flexural strengths, flexural toughness (residual strength), cracking resistance, and impact resistance of UHPC are mainly a consequence of the addition of fibers [21–23].

The fiber type affects the performance of the material with respect to these properties [24]. Fiber-reinforced concrete (FRC) has a pseudo-ductile behavior, with increased energy being necessary for failure of the tested element [25]. The addition of more than one type of fiber, whether by material or form, can result in gains in toughness and impact resistance [26].

A brittle matrix with fiber hybridization can behave differently from a composite with a mono-fiber type, which presents a simple crack when subjected to tensile stress [27]. As reported by Richardson et al. [26], the hybridization of steel fiber (90%) with polymeric fiber (10%) with a total amount of 1.15% can yield a greater level of hardness, toughness, and spalling resistance when used to test UHPC plate elements of 0.76 m in thickness subjected to impact loading of up to 6500 J. Máca et al. [28] evaluated the effect of fiber contents of 1%, 2%, and 3% by volume on the projectile impact behavior of UHPC with different chemical admixture and SCM contents. The authors concluded that the addition of 2% fiber presented the best performance for impact resistance. Wu et al. [29] found that cementitious matrices with high compressive strength made with at least 1.5% steel fiber could achieve adequate performance against ballistic impact energy levels greater than 4000 J. The structural behavior of UHPC elements subjected to ballistic impact was simulated using analytic computer-aided models [25]. The study indicated that the behavior of UHPC plate elements prepared with 2.5% steel fibers at thicknesses of 0.075 and 0.100 m showed satisfactory performance, with a low spalling index under projectile penetration loading.

The above studies demonstrate the advantages of using UHPC composites with fiber volumes between 1.5% and 3.0% to ensure high resistance to ballistic impact. Another justification of this study is the possibility of reducing the thickness of the walls used in ballistic protection systems to less than 70 mm with the use of UHPC. Normally, the thicknesses of conventional concrete walls/structures are over 150 mm to ensure safety on impact. Thus, the study reported here aims at developing thin UHPC plate elements with low cement contents (high volumes of SMC) that can lead to high impact strength and high energy dissipation for ballistic impact applications. Thin plate elements measuring 30, 50, and 70 mm in thickness were prepared with UHPC containing fiber volumes of 1.5% and 3.0%. UHPC hybrid fiber contents with 80% steel fibers and 20% polypropylene (by volume) were used to achieve enhanced mechanical performance and reduced costs. The evaluated properties included compressive strength, flexural tensile strength, residual flexural strength, and ballistic impact strength.

## 2. Experimental Procedure

### 2.1. Materials

Brazilian cement Type CP-II F40 was used along with fly ash (FA) and silica fume (SF), with densities of 2000 and 2200 kg/m$^3$, respectively. The Brunauer–Emmet–Teller (BET) surface areas of fly ash and silica fume were 28.380 and 20.000 m$^2$/kg, respectively. The BET surface area of cement was 471.5 m$^2$/kg. Quartz dust made of silicon dioxide (SiO$_2$) was used to replace some of the cement, with BET surface areas of 10.350 m$^2$/kg. The physical properties of the cement, SCMs, and quartz dust are presented in Table 1.

**Table 1.** Physical properties of SCMs.

| Materials | Specific Gravity (kg/m$^3$) | Bulk Density (kg/m$^3$) | BET (m$^2$/kg) |
|---|---|---|---|
| Cement | 0.31 | 0.14 | 471.5 |
| Fly ash | 0.20 | 0.12 | 28.380 |
| Silica fume | 0.22 | 0.04 | 20.000 |
| Quartz dust | 0.26 | 0.14 | 10.350 |

It should be noted that there is a need to develop a UHPC with low cement consumption. Thus, the binder was added by adding pozzolanic materials. Instead of adding just one type of SMC, several SCMs with different particle sizes were used to obtain an efficient particle size distribution.

Two types of natural sands (fine and medium sands with different grain size distributions) were used. Their unit weights were 1.420 and 1.540 kg/m$^3$, respectively, and their specific gravities were 2.600 and 2.610 kg/m$^3$, respectively. Figure 1 illustrates the combined particle size distribution of the cement, SMCs, quartz dust, and sands along with the A-A model curve that was used to enhance packing density, as in the methods reproduced in research by [30].

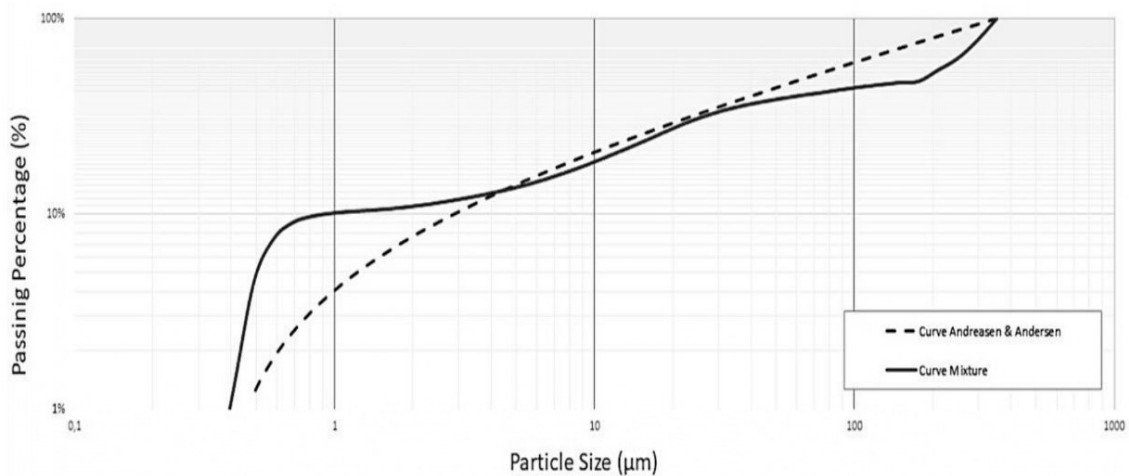

**Figure 1.** Particle size distribution curve of the UHPC studied in relation to the Andreasen–Andersen curve.

The UHPC mixtures were prepared with polycarboxylate-based superplasticizer and a viscosity-modifying admixture (VMA). The VMA used was Centrament Stabi 520. The superplasticizer had a pH of 6.9 and a specific gravity of 10.9 kg/m$^3$. The viscosity-modifying agent had a pH of 7.0 and a specific gravity of 10.0 kg/m$^3$.

Steel and polypropylene (PP) fibers were used in this study. The reinforcements were hybridized with 80% steel fibers and 20% PP fibers, as reported by Christ [18]. Hybrid fibers were used for cost reduction and enhanced mechanical performance of the UHPC. Table 2 summarizes the properties of these fibers.

**Table 2.** Physical and mechanical properties of fiber reinforcements.

| Property | Steel Fiber | PP Fiber |
|---|---|---|
| Diameter (μm) | 21 | 12 |
| Length (mm) | 13 | 6 |
| Tensile strength (MPa) | 2750 | 370 |
| Specific gravity (kg/m$^3$) | 7850 | 910 |
| Image | 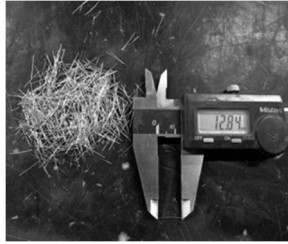 | 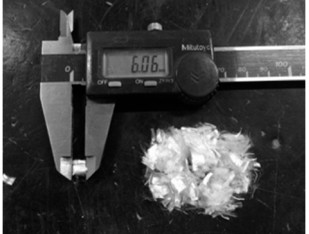 |

## 2.2. Mixture Design and Specimen Preparation

The UHPC mixture proportions are shown in Table 3. The mixture was performed in a vertical mixer with a capacity of 0.15 m$^3$. Silica fume, superplasticizer, and 80% of the total water were initially added and mixed for 8 min. Two sands were then added. Cement was added 2 min later, followed by viscosity-modifying agent (reducing the potential for segregation of the mixture), fly ash, quartz dust, and the remaining water in a 5-min timespan. Fiber reinforcements were added intermittently to avoid the formation of clusters.

**Table 3.** Mixture proportioning of UHPC made with 1.5% and 3.0% hybrid fiber contents.

| Materials | Proportion to Binder Content | Content (kg/m$^3$) |
|---|---|---|
| CPII-F40 | | 252 |
| Fly ash (FA) | 1 | 108 |
| Silica fume (SF) | | 214 |
| Quartz dust | | 292 |
| Medium sand | 2.39 | 583 |
| Fine sand | | 496 |
| Steel fiber (%) | 1.2% and 2.4% | 94.2 and 188.4 |
| PP fiber (%) | 0.3% and 0.6% | 2.7 and 5.5 |
| w/b | 0.25 | 143.5 |
| Superplasticizer (SP) | 22.9 L/m$^3$ | |
| VMA | 5.7 L/m$^3$ | |
| Air content | UHPC1.5 | 3.7% |
| | UHPC3.0 | 4.8% |

The investigated UHPC was prepared with a low cement content of 252 kg/m$^3$ and binder content of 574 kg/m$^3$ to enhance material unit cost. The cement content was much lower than the 889 kg/m$^3$ used by [31,32] and 450 kg/m$^3$ used by [33]. The composition of UHPC followed the parameters reported by [34]. The tested concrete plates and specimens were demolded 24 h after preparation and transferred to a curing room at a controlled temperature and humidity of 23 ± 2 °C and 98% ± 2%, respectively. The specimens were cured in these conditions up to 7 and 28 days, and the mechanical properties and projectile

impact characteristics were determined. The evaluated mechanical properties included uniaxial compressive strength, tensile strength, and flexural strength and toughness.

### 2.3. Experimental Methods

### 2.3.1. Mechanical Properties

A total of 3 cylindrical specimens measuring 50 × 100 mm were tested for axial compressive strength at 7 and 28 days. An Instron press was used to conduct this test at a loading rate of 0.45 MPa/s. The average of 3 specimens is reported. The flexural performance was determined according to [35] using a closed-loop system with controlled displacement of 0.002 mm/s. A universal electronic hydraulic testing machine with a capacity of 2 MN was used. UHPC prisms measuring 100 × 100 × 350 mm were loaded using 2 cleavers positioned on the middle third span of the specimen, as shown in Figure 2. The flexural load–displacement curve was recorded using 2 embedded linear variable differential transformers (LVDT) placed on both sides of the specimen. Two prisms per test were used for each level of fiber content. Flexural strength was calculated and 2 residual strengths of f100;0.50 and f100;2.0 at vertical displacements of 0.50 and 2.0 mm, respectively, were found.

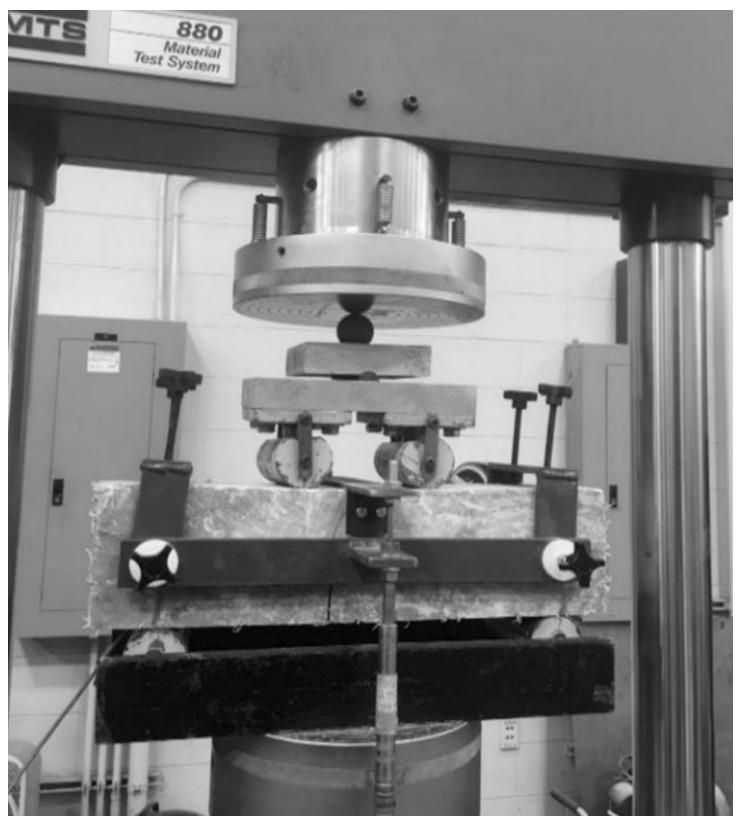

**Figure 2.** Illustration of 4-point bending testing for UHPC prisms.

### 2.3.2. Projectile Impact Resistance Testing

The projectile impact resistance of UHPC plate specimens was evaluated using projectiles shots ejected by guns with different calibers that could secure different impact loading rates. For each tested specimen, 5 impact loading rates were applied. This was achieved by changing the projectile mass. This was varied with the caliber of the bullets and the shooting distance from the beginning of the impact point, as illustrated in Figure 3. The shooting distance ranged from 5 to 15 m.

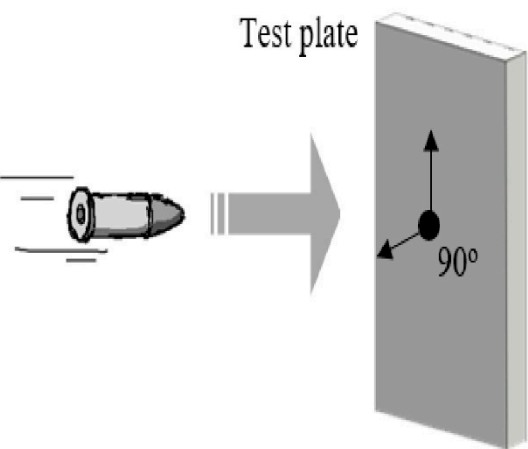

**Figure 3.** Illustration of motion and direction of projectiles shot against test plate sections.

Thin plate specimens measuring 500 × 500 mm were prepared with various thicknesses of 30, 50, and 70 mm. The ballistic impact strength test was performed to assess the strength of these elements by shooting 5 times (points) against each plate with 4 energies (4 caliber types), as depicted in Table 4. For each caliber, 6 thin plates were crafted. The impact energies were specified in accordance with the energies in other studies on ballistic impacts, as described by [36–38]

**Table 4.** Calibers and corresponding impact energies used.

| Caliber | | Impact Distance (m) | Impact Energy (J) |
|---|---|---|---|
| 380 SPL | | | 270 |
| 9 mm | | 5 | 540 |
| 12/70 | | | 4000 |
| 7.62 mm | | 15 | 7420 |

For each tested specimen, 5 impact loading rates were applied. This was achieved by changing the projectile mass with the caliber of the bullet and the shooting distance from the starting point of impact, according to [39].

The instrumentation for the impact tests counted with a special pin attached to the side and the base of each element to avoid tipping and other instabilities. Considering the steel frame and the cementitious sample, the stresses were absorbed only by the element. Thus, the sample was subjected to the impact of projectiles with known velocities on 5 points per element, as depicted in Figure 4. All tests were performed on UHPC specimens at 28 days, according to the requirements of [40,41]. Each plate was analyzed by measuring the depth of perforation made by the impact and the diameter of entry and exit of the projectile. The ballistic protective materials covered by [42] are classified into many types by level of performance, and these levels are indicated in Table 5.

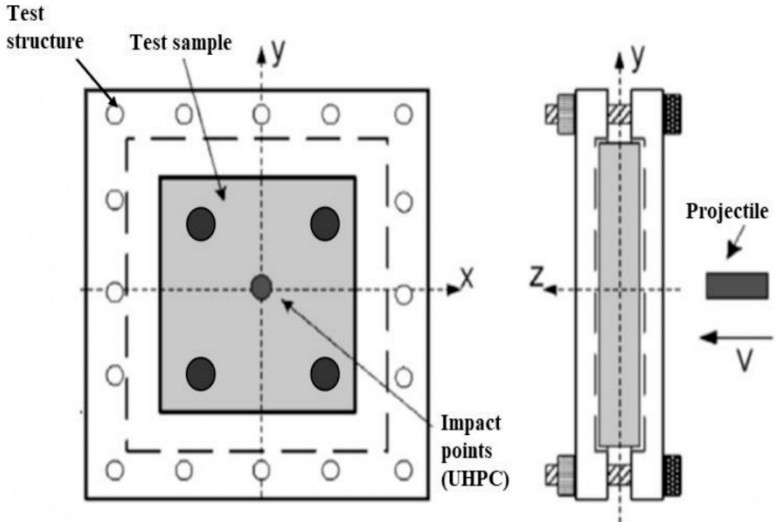

**Figure 4.** Test instrumentation marking the 5 impact points (Adapted from [42]).

**Table 5.** Levels of ballistic protection.

| Level | Caliber | Mass of Bullet (g) | $V_0$ (m/s) | Performance |
|-------|---------|--------------------|-------------|-------------|
| I | 0.38 Special RN [a] | $10.2 \pm 0.1$ | $254 \pm 15$ | NP [b] |
| II-A | Lower-velocity 357 Magnum | $8.0 \pm 0.1$ | $381 \pm 12$ | NP [b] |
| II | High-velocity 357 Magnum | $8.0 \pm 0.1$ | $425 \pm 15$ | NP [b] |
| III-A | 9 FMJ [c]<br>12-gauge rifle slug | $15.6 \pm 0.1$ | $426 \pm 15$ | NP [b] |
| III | $7.62 \times 51$ FMJ [c] | $9.7 \pm 0.1$ | $838 \pm 15$ | NP [b] |

[a] RN—round nose. [b] NP—no penetration. [c] FMJ—full metal jacket.

## 3. Results

### 3.1. Compressive Strength and Flexural Behavior

Table 6 summarizes the results for the compressive strength of UHPC specimens at 7 and 28 days. It was observed that the compressive strength of UHPC specimens at 7 days was above 70 MPa. Such behavior was expected due to the type of cement, high packing of particles, low water/cement, and proper curing. There was a small reduction in compressive strength for the mixture prepared with 3.0% of fibers in comparison with that made with 1.5% fiber content. The reduction was smaller than the standard deviation. According to Wang and Gao, a higher content of fibers can increase entrapped air voids (3.5–5.5%), hence impairing mechanical properties [42]. This performance was discussed in other publications [18–27]. It was stated that there was a direct relationship between

the fiber volume ratio and the mechanical strength of concrete [28]. Furthermore, Table 6 presents the flexural and residual strength results of the UHPC composites at 28 days.

**Table 6.** Compressive strength and calculated flexural and residual strengths of UHPC.

| Composite | 7-Day | 28-Day | | | |
|---|---|---|---|---|---|
| | $f'_c$ (MPa) | $f'_c$ (MPa) | $f_t$ (MPa) | $f_{100;0.50}$ (MPa) | $f_{100;2.0}$ (MPa) |
| UHPC1.5 | 72.9 | 104.8 | 12.5 | 10.8 | 12.5 |
| UHPC3.0 | 71.2 | 102.9 | 16.3 | 9.9 | 16.8 |

Wu et al. [29] developed UHPC made with 1.5% steel fibers with (28-day) compressive strength of 90 MPa that could show adequate ballistic impact performance. It should be noted that the hybridization of rigid fibers (steel) with bendable ones (PP) affects the compressive strength and elastic modulus of the composite according to Christ [18] and Ehrenbring et al. [43]. Since PP fibers are deformable and have relatively low stiffness, they end up occupying the physical space of materials with higher strength and stiffness of aggregate and other solid particles [44]. Thus, the compressible material is removed, and another material without these characteristics can take its place, hence generating internal fail points in the system and reducing the load-bearing capacity.

At 28 days, the compressive strength of the tested UHPC mixtures surpassed 100 MPa. The UHPC3.0 specimens achieved an average compressive strength of 102.9 MPa. As stated in the description of the composition, the use of pozzolanic materials was crucial to achieve the values found since the potential mechanical performance remains possible at more advanced ages due to the pozzolanic reaction. Moreover, both mixtures made with low cement content of 252 kg/m$^3$ achieved 100 MPa compressive strength, demonstrating that it is possible to secure adequate strength despite the low cement content.

The Table 6 reports the values of the residual strength (f100) at two different deflections. The first value corresponds to the load corresponding to a vertical displacement of 0.50 mm (f100;0.5), which can be considered as an average deformation in the elastic regime. The other corresponds to the residual strength at a deflection of 2.0 mm (f100;0.5), which corresponds to the ultimate deformation. Contrary to the compressive strength, the increase in fiber ratio is shown to improve flexural strength due to combined effect of anchoring and bridging. This can restrict matrix crack propagation because of the stress transferred from the matrix to the fiber at the interface, as reported by Figueiredo [23] and Christ [18]. Besides, increasing the content of fibers resulted in an increase in residual strength. For the UHPC1.5 specimens, the mean value of f100;2.0 was 12.5 MPa, while the UHPC3.0 specimens yielded a mean value of 16.3 MPa, which is 3 0.5% greater.

Figure 5 shows the flexural load–displacement curves of UHPC1.5 and UHPC3.0 specimens. The flexural load increased linearly with the increase in deflection up to approximately 0.5 mm. The load then increased non-lineally with deflection at a decreasing slope, and the specimens started to show multiple cracks. After its peak load, the load gradually decreased with deflection. The UHPC3.0 mixture exhibited greater peak load and energy dissipation (which is considered as the area under the load–displacement curve) as compared to the UHPC 1.5 mixture.

Both UHPC mixtures made with 1.5% and 3.0% fibers had similar residual strength at 0.5 mm displacement, differing only by 0.9 MPa. Such close values occurred mainly because of the similar matrix characteristics and the fact that fibers do not bear significant stress before the occurrence of cracking, as stated by Quinino [27]. In the initial stage, the matrix resists tensile stress and the internal reinforcement is not triggered. The fibers are solicited following the initiation and propagation of cracks and increased displacement.

The final residual strength of the samples was attained at a displacement of 2.0 mm and was higher for UHPC3.0. Such behavior is justified by the content of fibers added in the matrix. The fiber volume in the UHPC3.0 mixture was twice that in the UHPC1.5 mixture, hence increasing the energy required to lead to failure of the specimens. The

residual strength at 2.0 mm deflection reached 16.8 MPa, which was a value 35% higher than that of the UHPC1.5 mixture. The results are in accordance with findings of Ren et al. [45] who reported that fibers were more effective for increasing the amount of energy required to break these reinforced matrices. The disparity between residual strength values in this stage was due to physical and morphological modifications of the material during the tests, which triggered the reinforcement wholly at the fractured section.

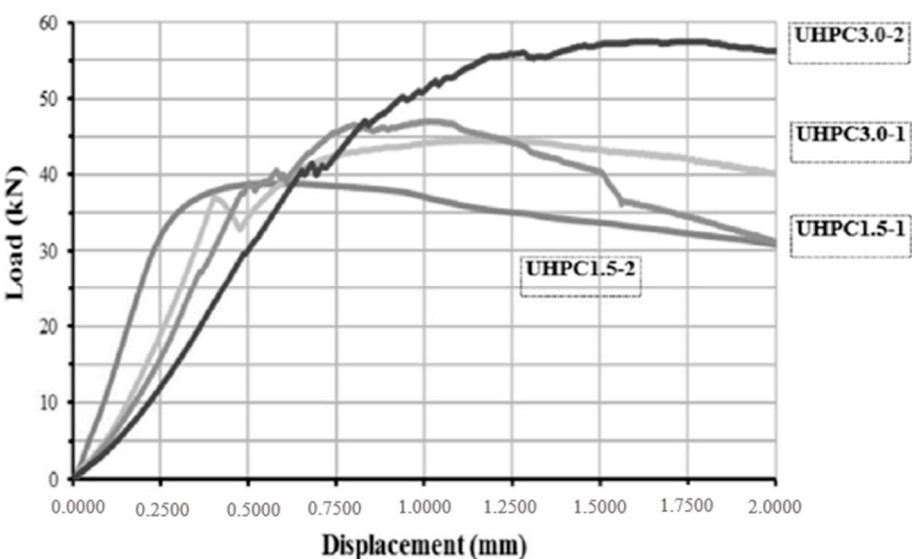

**Figure 5.** Flexural load–displacement curves of UHPC1.5 and UHPC3.0 made with hybrid fibers.

### 3.2. Resistance to Projectile Impact

Figure 6 presents the average results of the 5 points for projectile impact in the thin UHPC plates at 28 days. It shows how the different thin UHPC plates behaved with respect to the depth of penetration of the projectiles, considering their respective impact energies. For energies below 1000 J, the projectiles did not achieve depths greater than 0.010 mm. As for the impacts of 270 J, there was no penetration on any element, hence signalizing satisfactory behavior for low calibers. However, doubling the energy (540 J) broadened the damage since specimens with 1.5% fibers achieved an average depth of penetration of 6 mm. When the reinforcement ratio was increased to 3.0%, the projectile penetration was reduced by 25% and reached an average value of 0.0045 m.

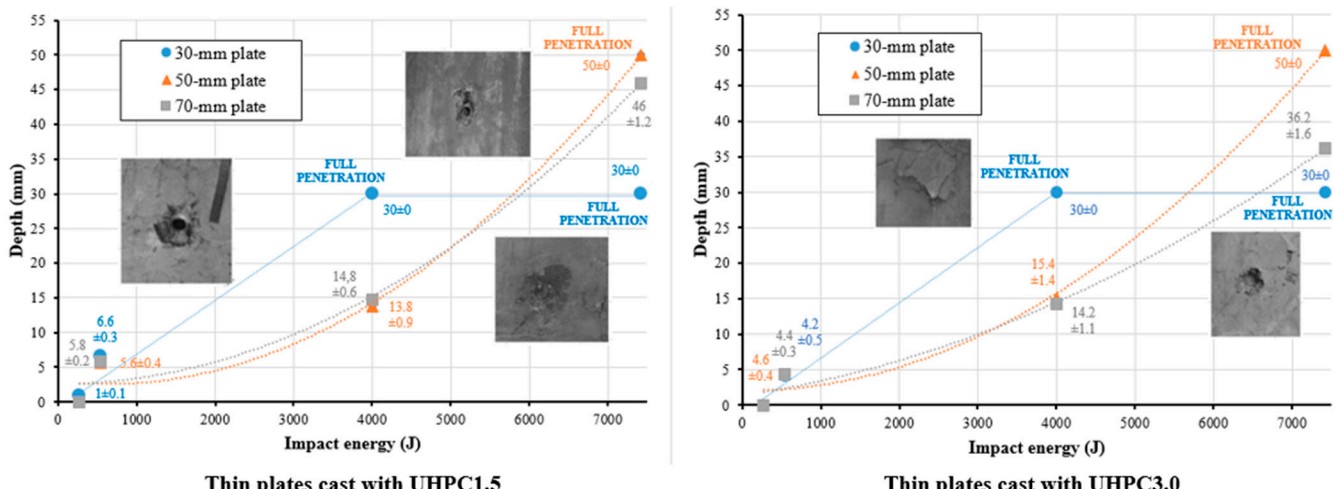

**Figure 6.** Depths of fracture in UHPC plates.

The geometric characteristics of the thin plates became more influential as the impact energies surpassed 1000 J. The 30-mm-thick specimens made with either of the fiber reinforcement ratios exhibited a projectile pass-through type of penetration starting at impact energy level of 4000 J. Owing to this impact energy, the level of fiber reinforcement and thickness of the test sample were not sufficient to fend off the stresses generated by the projectile, and the samples broke immediately. In the case of the 0.050 m plate elements, the average depth of penetration of the projectile for both fiber composites was 15 mm at 4000 J impact energy. Therefore, increasing the thickness of the plates by 66% reduced the extent of the damage by 100%. The increase in impact energy to 7000 J led to complete perforation of the projectile in the 50-mm-thick elements regardless of the fiber content.

The elements measuring 70 mm in thickness behaved in a similar way to those with thickness of 50 mm. Contrary to the 30- and 50-mm-thick elements, the projectile pass-through did not take place at any of the tested impact energies. For the UHPC1.5 specimens, the projectiles attained a depth of 45 mm. As for UHPC3.0, the depth was of 35 mm, which corresponds to 50% of the element's thickness. By doubling the fiber volume reinforcement to 3.0%, it was possible to reduce the damage caused by the 7.62 mm caliber bullet by 23%.

The above results demonstrate the influence of the plate geometric stiffness on the behavior resistance to projectile impact, which is due to the increased thickness of the element as analyzed with regard to the different impact energies. The trend line of plates with a thickness of 30 mm behaved inversely to the other sample regardless of the amount of fiber reinforcement. The 50- and 70-mm-thick plates behaved in a similar manner in terms of the differences in penetration depth. The increase in impact energy affected the penetration depth of the projectiles. The geometric stiffness of the piece increased because of the increase in thicknesses, which is related to gain in inertia (I). Thus, thicker samples achieved smaller penetration depths regardless of the fiber content. The increase in fiber volume from 1.5% to 3.0% did not improve the impact penetration resistance, although gains were noted for the 70-mm-thick plate test. This contradiction of results correlates to the number of fibers found in the breaking section of the test specimens. For instance, the thinner plates tended to contain less fibers in their cross-sections than the 70-mm-thick samples. Therefore, specimens with larger thicknesses presented higher content of fiber-reinforced points in the fragmented section, thus averting crack propagation while increasing the absorption of impact energy from the projectiles.

Figure 7 presents the spalling volume of the UHPC1.5 and UHPC3.0 plates as a function of impact energy at 28 days. It demonstrates the sensitivity of 0.030-m-thick plates to spalling volume. At 4000 J of impact energy, the 0.030-m-thick samples yielded the highest spalling level of 240 cm$^3$. The results are related not only to the increased energy but also to the type of projectile in use. The bullet caliber leads the projectile to shatter into small fragments, thus increasing the area of impact subjected to the energy of 4000 J. Compared to the higher energy of 7420 J, spalling remained below 100 cm$^3$ since the impact from the projectile caused damage. For all test samples, the spalling results for energies below 1000 J were similar and remained below 10 cm$^3$. At the 4000 J impact energy, the fiber ratio mainly influenced the spalling volume for thinner plates. Doubling the fiber reinforcements reduced the spalling volume by 42%. Therefore, the thicker the plate is, the smaller the effect of the fiber ratio becomes.

For impact energies of 7420 J, due to their small thicknesses, the 30-mm-thick plates underwent less spalling than the 50-mm-thick plates. This is linked to the absorption of potential ballistic impact energy. As the impact energy was much higher than the geometric stiffness of the plate, the projectile pierced without causing expressive spalling. Nevertheless, the impact energy of the 50-mm-thick plates caused more damage due to the reinforcement being more balanced and rendering a broader area of energy dissipation. The spalling volume of elements of 50 mm remained above 260 cm$^3$, which is the highest value obtained in this research. In that case, the increased fiber content did not provide benefits to the system.

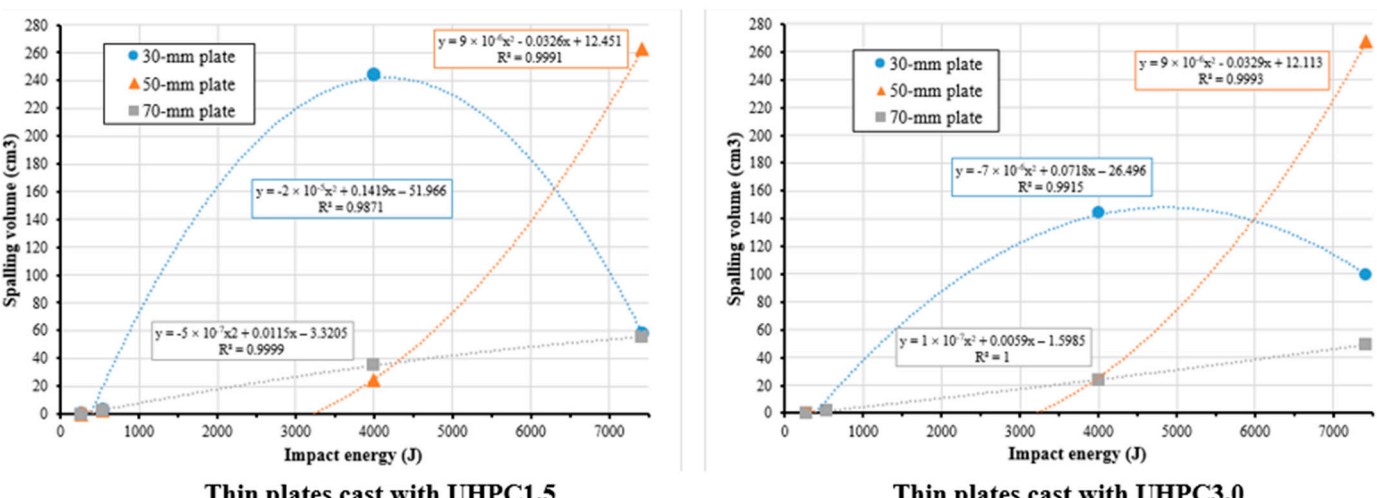

**Figure 7.** Spalling volume for each thickness of the UHPC plates.

The 70-mm-thick samples revealed the smallest spalling volume. As the impact energies increased, the spalling varied as well. However, no projectile passed through the elements, suggesting that the system is adequate for impacts of multiple magnitudes. Moreover, for thicker elements, the fiber ratio did not yield significant performance gains when considering that the increased stiffness became a more efficient measurement. The increased volume of reinforcement no longer had the primary influence on performance, resulting in increased costs and less workable mixtures.

Figure 8 depicts the unit values and the efficiency of both fiber volume ratios with regard to the spalling/impact energy function. Differences in performance can be noted up to an energy level of 4000 J. These differences can reach up to 40% between the UHPC1.5 and UHPC3.0 mixtures. Thus, the viability of adding more reinforcement to reach better performance was assessed. The increased fiber ratio seemed to be inefficient at high energy levels. Therefore, the elements' geometric characteristics (thickness) should be prioritized, as depicted in Figure 7.

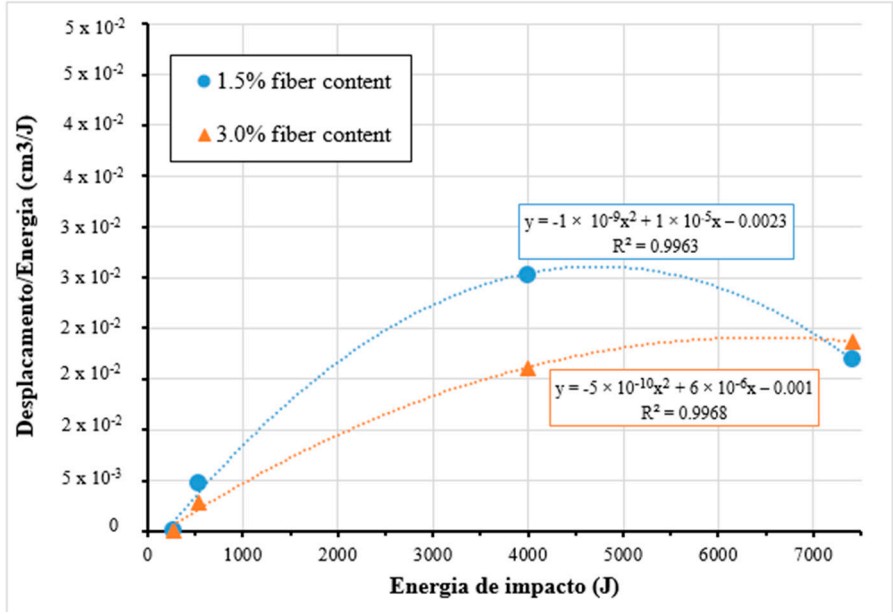

**Figure 8.** Efficiency of reinforcements comparing spalling volume over impact energy.

Table 7 classifies the plates according to the ballistic protection level [46] that can be achieved at different impact energies. Thus, Type I refers to protection (no penetration) against the standard test impact of most handgun rounds with calibers of 38, 32, and 25 mm. Type II-A refers to protection (no penetration) against the standard test impact of handgun rounds with a caliber of 9 mm and the lower-velocity 357 Magnum. Type II refers to protection against handgun rounds with a caliber of 9 mm and the high-velocity 357 Magnum. Type III-A refers to protection (no penetration) against standard test impact rounds from a medium-powered rifle (9 FMJ and 12-gauge rifle slug). Finally, Type III refers to protection against a high-powered rifle (7.62 mm).

As expected, samples with a thickness of 70 mm have the potential to reach higher ballistic protection levels (Type III) than those with thinner plates. In addition, only the 70 mm UHPC plates achieved the maximum ballistic protection for the 7.62 mm caliber bullet, whereas the other samples suffered projectile pass-through at the relatively high-test energy of 7420 J, which disqualifies them as elements of protection. Therefore, UHPC sections made with a thickness of 70 mm were found to be an interesting alternative due to their great performance during the impact testing, thus resulting in safer buildings.

**Table 7.** Classification of ballistic protection level of tested UHPC plates.

| Plate Thickness (mm) | Caliber and Impact Energy (J) | Fiber Volume (%) | Ballistic Protection Level |
|---|---|---|---|
| 30 | 0.380 SPL (270 J) | 1.5 | I |
| | | 3.0 | |
| | 9 mm (540 J) | 1.5 | II |
| | | 3.0 | |
| | 12/70 (4000 J) | 1.5 | FULL PENETRATION |
| | | 3.0 | |
| | 7.62 mm (7420 J) | 1.5 | |
| | | 3.0 | |
| 50 | 0.380 SPL (270 J) | 1.5 | I |
| | | 3.0 | |
| | 9 mm (540 J) | 1.5 | II |
| | | 3.0 | |
| | 12/70 (4000 J) | 1.5 | III-A |
| | | 3.0 | |
| | 7.62 mm (7420 J) | 1.5 | FULL PENETRATION |
| | | 3.0 | |
| 70 | 0.380 SPL (270 J) | 1.5 | I |
| | | 3.0 | |
| | 9 mm (540 J) | 1.5 | II |
| | | 3.0 | |
| | 12/70 (4000 J) | 1.5 | III-A |
| | | 3.0 | |
| | 7.62 mm (7420 J) | 1.5 | III |

## 4. Conclusions

Testing was carried out to evaluate the feasibility of producing UHPC elements with a relatively low compressive strength of 100 MPa prepared with low cement content and

hybrid fibers to enhance the ballistic protection of building structures. Based on the results of the testing program, the following conclusions can be drawn:

- Similar compressive strength was attained for UHPC mixtures made with hybrid fiber volumes of 1.5% and 3.0%. However, the latter mixture resulted in higher residual flexural strength. In addition, the flexural toughness of the UHPC3.0 mixture was 34.4% greater than that of the UHPC1.5 mixture, affecting the ballistic impact resistance of the material.
- Plate specimens measuring 70 mm in thickness made with either 1.5% or 3.0% fiber content could stop gunshots with impact energy up to 7420 J and ensure level-III ballistic protection.
- The highest spalling levels on the surface of impact were obtained with caliber 12/70 bullets that led to dispersion of the projectiles. However, the highest spalling levels on the surface opposed to the impact were obtained with caliber 7.62 mm bullets (7420 J) due to the absorption and dissipation of energy on the specimens.
- Increasing the thickness of the plates can increase the ballistic protection level of the elements and reduce the extent of spalling.
- The fiber ratio had no significant effect on spalling and penetration depth for the tested UHPC plate specimens, except for the occasion when an energy equivalent to 7420 J was attained.

**Author Contributions:** Development of the experimental tests and results compilation, P.R.D. and G.C.M.; Results analysis, discussion and comparisons with research sources, H.Z.E., F.P. and R.C.; Review of results and valuable contributions in their interpretation, M.F.d.O. and B.F.T. All authors have read and agreed to the published version of the manuscript.

**Funding:** This research received no external funding.

**Informed Consent Statement:** We declare consent.

**Data Availability Statement:** We declare data availability.

**Acknowledgments:** The authors are grateful for the assistance received in carrying out the tests and purchasing raw materials, especially from Itt Performance—UNISINOS.

**Conflicts of Interest:** We declare that there is no conflict of interest.

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
