# Peer review of "Ballistic Impact Resistance of UHPC Plates Made with Hybrid Fibers and Low Binder Content"

_sustainability, doi:10.3390/su132313410_

Round 1

Reviewer 1 Report

Dear authors, 
Thank you for the interesting work, however revisions are needed to improve the clarity of paper presentation. 

1. Avoid using abbreviations/acronyms in Title. 
2. What experimental methods were used? This should be clear in the summary.
3. Introduction lacks clarity add few more literatures from the host journal. In the introduction, you need to connect the state of the art to your paper goals. Please follow the literature review by a clear and concise state of the art analysis. You should clearly show the identified knowledge gaps and link them to your paper goals. Please reason both the novelty and the relevance of your paper goals, and define clearer the objective of your paper.
The last part of the introduction should conclude the limitations of the previous studies and provide the main objectives and novelties of this study in bullet points.

4. Please eliminate the use of redundant words. E.g. Respectively, therefore, thus, hence, finally, however, moreover, in addition. - Please revise all similar cases, as removing these term(s) would not significantly affect the meaning of the sentence. This will keep the manuscript as CONCISE as possible. Please check ALL.
Avoid beginning or end a sentence with one or a few words, they are usually redundant. Kindly revise all.

5. Please check the overall manuscript format for this journal (highlights, citation, labeling figures/tables, references, etc.). 

6. English - Please proofread the paper to improve the English standard for the entire manuscript. Spelling, punctuation, capitalization and spacing should also be revised intensively for a clearer presentation and sentence fluency. 

7. The conclusion should be majorly revised, and the main results should be summarized as the bullet points at the end of the conclusion. You should follow this structure: 1. A brief description of what have you done and what is your novelty? 2. How did you investigate the system and what are the main useful parameters? 3. Provide the primary results as the bullet points? 4. State the main limitations of this study and present some suggestions for future researches. 

Author Response

Thanks for the brilliant contributions. They were very important and of great value. Below we answer the questions and observations.

1. Avoid using abbreviations/acronyms in Title. Resp.: We understand the questioning, but the acronym "UHPC" is well established in academic and technical circles. As well as there are several articles that also use the term in the title.
2. What experimental methods were used? This should be clear in the summary. Resp.: It is described that an experimental evaluation was carried out and the tests carried out.
3. Introduction lacks clarity add few more literatures from the host journal. In the introduction, you need to connect the state of the art to your paper goals. Please follow the literature review by a clear and concise state of the art analysis. You should clearly show the identified knowledge gaps and link them to your paper goals. Please reason both the novelty and the relevance of your paper goals, and define clearer the objective of your paper.
The last part of the introduction should conclude the limitations of the previous studies and provide the main objectives and novelties of this study in bullet points. Resp.: Reviewed.

4. Please eliminate the use of redundant words. E.g. Respectively, therefore, thus, hence, finally, however, moreover, in addition. - Please revise all similar cases, as removing these term(s) would not significantly affect the meaning of the sentence. This will keep the manuscript as CONCISE as possible. Please check ALL.
Avoid beginning or end a sentence with one or a few words, they are usually redundant. Kindly revise all.  Resp.: Once the text has been revised, some suggested words are extremely necessary for understanding the text.

5. Please check the overall manuscript format for this journal (highlights, citation, labeling figures/tables, references, etc.).  Resp.: Reviewed.

6. English - Please proofread the paper to improve the English standard for the entire manuscript. Spelling, punctuation, capitalization and spacing should also be revised intensively for a clearer presentation and sentence fluency.  Resp.: Reviewed.

7. The conclusion should be majorly revised, and the main results should be summarized as the bullet points at the end of the conclusion. You should follow this structure: 1. A brief description of what have you done and what is your novelty? 2. How did you investigate the system and what are the main useful parameters? 3. Provide the primary results as the bullet points? 4. State the main limitations of this study and present some suggestions for future researches. Resp.: Reviewed. 

Reviewer 2 Report

See attached mandatory corrections

Author Response

Thanks for the contributions, some of the suggested articles were placed in the document. And the other contributions were reviewed and revised.

Reviewer 3 Report

The paper present an interesting study regarding the performance of thin plates of UHPC made using low cement content and steel and PP fibers. It can be very useful in designing buildings that need protection against explosion and impact. Also, replacing some of the classic materials used in concrete, a lower environmental impact can be achieved.

The introductory part is well described, the necessity of the study is argued. The materials used can be described better, specifying the standards used, the experimental methods are well explained, the results are inline with others mentioned in the literature. The conclusions can be more detailed, adding some personal opinions, not only presenting the obtained results that were already described in a previous chapters.

But, there are way to many grammar and spelling mistakes, paragraphs that need revisions, values improper used, wrong cited references, etc... all these make the paper very hard to read and understand. Below there are just a few observations that need to be addressed in order to help readers in a correct understanding of the presentation:

  • all references in the text should be revised using the journal's template, like [1], [2-4] etc; line 35 - there is an error showing a missing reference (4); it seems that not all the references are present in the text...
  • line 53 - use "according" to [15], not "to 15"; revise all paper;
  • line 94 - the plates' thickness is 30, 50 and 70 mm, not 3, 5 and 7 mm...
  • many errors in writing the correct values of  different parameters; see chapter 2.1 and table 1: the BET for cement is 471.5 m2/kg in the text and 0.4715 m2/kg in the table; also, for fly ash is 20,000 in the text and 28.3800 in the table; similar for silica fume and quartz dust...
  • the above values are 20.000 (20) or 20,000 (20 thousand)? pay attention to the use of "." and "," for decimal values...
  • lines 96, 125 - replace "to cost reduction" with "for cost reduction" or "to reduce costs"; 
  • line 109... the binder was added by adding...?
  • lines 113 - 114 - specific gravity for sand can not be 26 kg/m3;
  • how many specimens were tested for compressive strength?
  • why 50x100 mm for cylindrical specimens and 100x100x350 for prism? these values are according to Brazilian standards? also, the number of tests should be minimum 6, not 4...
  • line 173 - concrete plates can not be 500 x 500 m!
  • line 182 - figure 4, not 2;
  • line 182 - according to 35, line 193 - adapted from 38? for the same figure?
  • revise the caption for table 5... "a" is for "RN", etc
  • line 200 need to be rephrased;
  • values in table 6 are in Pa or in MPa? rearrange the table' cells so that it can be understand correctly... 28-day is outside its values;
  • figure 5 - what is UHPC3.0-2?
  • etc... to many ...

A major revision regarding the English language is recommended.

Author Response

We thank the reviewer for his brilliant contributions. And we answer them all below.

  • all references in the text should be revised using the journal's template, like [1], [2-4] etc; line 35 - there is an error showing a missing reference (4); it seems that not all the references are present in the text... Resp.: All points have been revised.
  • line 53 - use "according" to [15], not "to 15"; revise all paper; Resp.: Revised.
  • line 94 - the plates' thickness is 30, 50 and 70 mm, not 3, 5 and 7 mm... Resp.: Revised.
  • many errors in writing the correct values of  different parameters; see chapter 2.1 and table 1: the BET for cement is 471.5 m2/kg in the text and 0.4715 m2/kg in the table; also, for fly ash is 20,000 in the text and 28.3800 in the table; similar for silica fume and quartz dust... Resp.: Revised.
  • the above values are 20.000 (20) or 20,000 (20 thousand)? pay attention to the use of "." and "," for decimal values... Resp.: Revised.
  • lines 96, 125 - replace "to cost reduction" with "for cost reduction" or "to reduce costs"; Resp.: Revised.
  • line 109... the binder was added by adding...? Resp.: we don't understand the question.
  • lines 113 - 114 - specific gravity for sand can not be 26 kg/m3; Resp.: Revised.
  • how many specimens were tested for compressive strength? Resp.: 3 specimens were tested for each sample.
  • why 50x100 mm for cylindrical specimens and 100x100x350 for prism? Resp.: International and Brazilian standards allow the evaluation of compressive strength with different dimensions of specimens, as long as the relationship between diameter and height is equal to 2. these values are according to Brazilian standards? also, the number of tests should be minimum 6, not 4...  Resp.: The minimum number of specimens is 2 for compression and 5 for flexion traction. Only two prismatic specimens were used for each sample. This was a limitation of (study), and informed in the text.
  • line 173 - concrete plates can not be 500 x 500 m! Resp.: Revised.
  • line 182 - figure 4, not 2; Resp.: Revised.
  • line 182 - according to 35, line 193 - adapted from 38? for the same figure? Resp.: Revised.
  • revise the caption for table 5... "a" is for "RN", etc Resp.: Revised.
  • line 200 need to be rephrased; Resp.: Revised.
  • values in table 6 are in Pa or in MPa? rearrange the table' cells so that it can be understand correctly... 28-day is outside its values; Resp.: Revised.
  • figure 5 - what is UHPC3.0-2? Resp.: It is the second prism in the sample with 3% fiber.
  • etc... to many ... Resp.: Revised.

A major revision regarding the English language is recommended. Resp.: Revised.

Round 2

Reviewer 3 Report

The authors highly succeeded in improving the paper in all the aspects suggested by the reviewers. The revised version is now acceptable for publishing.